# Thermomechanical Characterization and Modeling of Cold-Drawing of Poly(ethylene Terephthalate)

**DOI:** 10.3390/polym11111871

**Published:** 2019-11-13

**Authors:** Jürgen Oberer, Konrad Schneider, Jens-Peter Majschak

**Affiliations:** 1Technische Universität Dresden, Institute of Natural Materials Technology, Bergstr. 120, D-01069 Dresden, Germany; jens-peter.majschak@tu-dresden.de; 2Leibniz-Institut für Polymerforschung Dresden, Hohe Straße 6, D-01069 Dresden, Germany; schneider@ipfdd.de

**Keywords:** polyethylene terephthalate, strain-induced crystallization, strain rate, IR thermography, cold drawing, cold crystallization, glass transition, semi-crystalline, draw ratio, thermomechanical model

## Abstract

The tensile testing of amorphous polyethylene terephthalate is observed until failure by IR thermography and optical strain measurement. The deformation can be subdivided in six deformation phases: elastic deformation, neck formation with a localized sharp temperature rise, neck propagation, which is also known as cold-drawing, with heat generation in a transition zone, crack initialization with local heating, crack growth, and rupture. These deformation phases are showing different mechanical and thermal reactions to the deformation. The initial and drawn samples are studied with differential scanning calorimetry. Alongside heating due to the dissipation of mechanical energy, latent heat due to strain-induced crystallization was detected. While the material is cold-drawn, a high dependence on the crosshead speed is found for the heat generation as well as the draw ratio, mechanical response, and morphological changes due to orientation and crystallization. For cold-drawing, a thermomechanical model is introduced, which is based on the first law of thermodynamics and reproduces the temperature distribution along the sample.

## 1. Introduction

The dissipation of energy and heating of matter due to plastic deformation is well known and accepted as a pivotal mechanism in the deformation and failure of thermoplastic polymers. Many publications describe self-heating effects for cold drawing and the necking phenomenon [1,2,3,4,5,6,7,8,9,10]. Haward gives a survey of different heating effects in thermoplastics due to deformation. It is reported the temperature rise and heat generation is at its highest in the first occurrence of the neck [9]. The actual values for the maximum temperature differ for the different publications, as they are highly dependent on the polymer and the method of measurement. Bazhenov tried using powder with different melting points to determine the actual temperature and found the temperature in cold-drawing of polyethylene terephthalate (PET) films to be above 140 °C for a crosshead speed of 1000 mm/min [7]. 

In contrast, Liao et al. determined the maximum temperature for the cold-drawing of fibers of PET with infrared thermography to be about 85 °C for a crosshead speed of 750 mm/min [8]. Most authors used thermography systems to determine the temperature evolution, with one of the first reported instances found from Cross et al. [11] A dependence of the phenomena on the deformation rate is widely acknowledged. Bazhenov found for PET films a dependence of the engineering stress during neck propagation on the crosshead speed [7]. Liao and Meng found a dependence of the temperature rise in the transition zone in experiments on polypropylene (PP) and PET fibers, with a greater temperature rise in PET than polypropylene (PP) [4,12]. As an explanation for the significant temperature rise, Liao gives various reasons, including heat dissipation by viscous friction and latent heat developed by crystallization. The orientation and crystallization of the material caused by the stretching leads to a significant increase in the stiffness and toughness of the drawing direction, which has been used in film stretching and especially the drawing of fibers [4,5,13,14].

Bazhenov introduced a mathematical description of the adiabatic temperature development, based on the work of Toda, and extended it later for non-adiabatic cases, which occur at low drawing speeds. As the heat source, Bazhenov calculated the work performed on the sample and multiplied it with a factor that was determined experimentally [2,6,7]. Liao et al. emphasized the influence of the morphologic evolution due to the orientation and strain-induced crystallization of cold-drawn PET and PP [8].

In this publication, an overview of the tensile deformation of amorphous PET (APET) is given, until failure of the sample under consideration of the deformation-induced heating of the sample. This self-heating can be correlated with the stress evolution and be used to differentiate deformation phases. The temperature is measured by infrared thermography. Additionally, differential scanning calorimetry (DSC) measurements were used to determine the structural changes in the stretched material. Furthermore, a model for the temperature evolution in cold-drawing, which is based on the first law of thermodynamics, is introduced. Compared to Bazhenov’s approach, the dissipation of mechanical work and latent heat due to morphological evolution in the material are used as sources for heating the polymer.

## 2. Thermomechanical Model

A model is derived to describe the temperature evolution in the specimen due to the neck propagation. This model describes the temperature evolution dependent on time and position in the specimen, as well as the propagation of the neck in the specimen. As the energy balance states, the work $W_{12}$ performed on a system and the heat exchange $Q_{12}\;$ through to borders cause a change in the internal energy, which is represented by ΔU:(1)$$Q_{12}+W_{12}=\Delta U$$

Relevant contributions for changes in the internal energy for the given system regarding the drawing of APET are the change in heat content $mc_{p}\Delta T$, where m is the mass of the system, $c_{p}$ is the specific heat capacity, and ΔT is the temperature change. Furthermore, the evolution of stored crystallization enthalpy ΔH and the change in the stored elastic energy $\Delta E_{el}$ contribute to the changes in internal energy:(2)$$\Delta U=mc_{p}\Delta T+\Delta H+\Delta E_{el}$$

The performed work on the global specimen for uniaxial tensile tests is the integral of force over the displacement:(3)$$W_{12}=\int F\;dx$$

The work performed on a volume element dV, can be calculated by stress σ and strain ε values as:(4)$$dW_{12}=dV\int \sigma \;d\varepsilon$$

In this way, the performed work can be subdivided in a fraction, which is stored in the material as elastic energy $E_{el}$, and another fraction of work, which causes an irreversible deformation and dissipates in heat through internal friction [12].

The heat exchange $Q_{12}$ consists of the heat transfer in the specimen by heat conduction ${\dot{Q}}_{conduction}$ and the heat flux to the ambience ${\dot{Q}}_{ambience}$, which incorporates convection and heat radiation.

(5)$${\dot{Q}}_{12}={\dot{Q}}_{conduction}+{\dot{Q}}_{ambience}$$

For the transient heat conduction, Equation (6) is used:(6)$${\dot{Q}}_{conduction}=-\lambda \;\nabla^{2}\;T\;dV$$
with the assumption of a constant coefficient for heat conduction λ in a volume element dV. The heat flux to the ambience ${\dot{Q}}_{ambience}$, is given as:(7)$${\dot{Q}}_{ambience}=\alpha \;A_{S}\;\left(T-T_{ambience}\right)+\varepsilon_{radiation}\;\sigma_{S}\;A_{S}\;\left(T^{4}-T_{ambience}^{4}\right)$$
with the heat transfer coefficient α for convection, the emissivity $\varepsilon_{radiation}$, and the Stefan–Boltzmann constant $\sigma_{S}$ for heat radiation. $dA_{S}$ gives the free differential surface of the specimen, $T_{ambience}$ is the ambient temperature, and T is the temperature of the volume element.

When Equations (2)–(7) are inserted in Equation (1), with use of the specific values and some changes are done, one gets for the temperature change ΔT:(8)$$\Delta T=\frac{1}{c_{p}}dh+\frac{1}{c_{p}\rho }\int \sigma d\varepsilon -\left(\frac{\lambda }{c_{p}\rho }\nabla^{2}\;T\;dt\right)\;-\left(\frac{1}{c_{p}\rho }\frac{A_{S}}{dV}\left(\alpha \left(T-T_{ambience}\right)+\varepsilon_{radiation}\;\sigma_{S}\;\left(T^{4}-T_{ambience}^{4}\right)\right)dt\right)$$

In Equation (8), the first and second terms can be considered as the heating source, which consists of the change in the die-specific latent heat and the dissipated work. The third term considers the heat conduction in the specimen, and the fourth gives the heat transfer to the ambience. This formulation can be used universally for all deformation-induced heating, with some adaptions according to each case.

## 3. Materials and Methods

### 3.1. Materials and Sample Preparation

The specimen used in this study was cut off from a commercial film of amorphous PET of 400-µm thickness, as it is used for thermoforming, manufactured by WIPAK, Waldsrode, Germany. The glass transition in the initial state is at about 76 °C, and the density amounts to 1.38 g/cm³. The dog-bone specimens had a gauge length (parallel sample length) of 10 mm, in the cross-direction of the film, and a width of 4 mm with a radius of 5 mm for the transition from the gauge length to the gripping section, as shown in Figure 1, on the right.

To investigate cold-drawing, the specimen was modified. Coming from one side of the gauge length, there were two markings for optical strain measurement with a distance of 5 mm and on the other side, the specimen was kinked to give a predetermined point for initial necking, and the neck propagates through the whole gauge length.

### 3.2. Mechanical and Thermographic Studies

The specimens were stretched by 50 mm, or until break at different crosshead speeds on a tensile machine (AllroundLine Z020, ZwickRoell, Ulm, Germany). The temperature and strain were simultaneously monitored with the thermographic system ImageIR^®^ 8300 by InfraTec GmbH, Dresden, Germany, and the optical camera system MotionBLITZ^®^ CUBE7 by Microtron GmbH, Unterschleissheim, Germany. Figure 1 gives the experimental setup. The thermographic system is arranged normal to the sample. For the thermographic measurements, calibration curves were determined for a temperature range from 20 °C to 90 °C. From these curves, an emissivity $\varepsilon_{radiation}=0.95\;\left(\pm 0.015\right)$ was determined.

The optical system is put besides the IR camera, so that the optical axis has an angular offset to the normal direction. The horizontal distortion can be neglected as the strain is monitored only in the tensile direction. For better contrast in the optical measurement, diffuse backlighting is used through a screen behind the specimen. The backlighting is out of the image area of the thermographic system due to the angular offset, so there is no influence on the thermography.

### 3.3. Differential Scanning Calorimetry

DSC measurement is realized with a DSC Q 2000 from TA Instruments, New Castle, DE, USA, from −80 °C up to 300 °C at a heating and cooling rate of 10 K/min. Specimens of the initial material and the cold-drawn specimens with different crosshead speeds of about 4 mg were placed in standard aluminum pans.

### 3.4. Application of Thermomechanical Model for Cold-Drawing

For cold-drawing, some adaptions and simplifications to the model given above can be made. With the assumption of a uniform temperature distribution over the specimen width and because the specimen can be treated as a thermal thin body in the thickness direction, a one-dimensional approach for heat conduction is chosen:(9)$$\Delta T=\frac{1}{c_{p}}dh+\frac{1}{c_{p}\rho }\int \sigma d\varepsilon -\left(\frac{\lambda }{c_{p}\rho }\frac{\partial^{2}T}{\partial x^{2}}dt\right)\;-\left(\frac{1}{c_{p}\rho }\frac{A_{S}}{dV}\left(\alpha \left(T-T_{ambience}\right)+\varepsilon_{radiation}\;\sigma_{S}\;\left(T^{4}-T_{ambience}^{4}\right)\right)dt\right)$$

The Biot number helps classify a thermodynamic system in a thermal thin body with small Biot numbers, where heat conduction can be neglected or result in a thermal thick body. With a coefficient of heat conduction $\lambda =0.25\frac{W}{m\;K}$, which is taken from Bazhenov [7], a heat transfer coefficient of $\alpha =55\frac{W}{m^{2}\;K}$, which is determined through free cooling curves according to Newton’s law of cooling, and a specimen thickness $h_{film}=400\;µm$, the Biot number:(10)$$\mathrm{Bi}=\frac{\alpha \;h_{film}}{2\;\lambda }=0.044$$
is calculated. Heat transfer is highly dependent on the airflow and thus from the experimental setup, but the given value seems to be valid for different crosshead speeds. In consideration of this, as well as the stretched area of the specimen is used for the cooling curves, where there is no more deformation and heat generation. For convective systems, an analysis with an approximation for various shapes with Bi = 0.1 showed that the largest temperature deviation is within 2% of the mean temperature difference, so the assumption of a thermal thin body is appropriate [15]. Changes in the density, heat capacity, and thermal conductivity due to temperature dependence or a change in the morphology are neglected, as the effects are estimated to be small and with no effect on the qualitative trend of the model.

## 4. Results

### 4.1. Experimental Overview

A typical engineering stress curve for polymers with necking behavior is given until failure in Figure 2 as well as the corresponding evolution of the maximum temperature in the sample. Engineering stress and maximum temperature are not based on engineering strain as customary, but rather on time, as strain is found to be delusive for experiments where the specimen develops areas with a high drawing ratio on one hand and on the other hand, keeps areas with mostly elastic deformation.

The deformation is classified in six deformation phases; see also Grellmann et al. [14]. Phase I characterizes the mostly elastic deformation where the thermoelastic effect can be observed with a marginal decrease of temperature with increasing strain. Phase II shows the neck formation with the typical decrease of engineering stress but additionally—as shown in Figure 2 in the diagram and the corresponding picture A for phase II—a great rise and a following drop in local temperature. Phase III is the neck propagation with constant levels of engineering stress as well as a relatively constant maximum temperature. A transition zone is formed on one side of the neck in which the plastic deformation and heat generation is localized, which can be seen in Figure 3. In this transition zone, the material is stretched with remarking heat production as the neck propagates in the specimen. In the areas outside the transition zone, stretched or unstretched, there seems to be no heat generation. If one runs on the temperature profile in Figure 3 from left to right, it equals the Eulerian way of thinking for one point that is overrun by the propagating neck. Starting on the left side, one can see some heating of the unstretched area due to heat conduction from the high-temperature gradient to the transition zone, which is marked as the area with the highest temperature. On the other side of the transition zone, one sees a smooth cooling curve until the temperature drops to ambient temperature. This cooling curve can be used to determine the heat transfer coefficient α, as mentioned above. In phase IV (Figure 2), a crack is formed in the specimen. One can see a rise in the maximum temperature that corresponds to a strong localized heating at the left edge of the specimen, as one can see in picture C, Figure 2. This heating is most likely caused by the sliding of the polymer chains with respect to each other. The engineering stress shows spontaneous decline.

For phase V, the maximum temperature in Figure 2 shows a clear differentiation to phase IV as a small temperature peak, which can be interpreted as a local rupture, which is followed by crack growth with more or less a constant level of maximum temperature in the crack tip and the edges, see picture D, Figure 2. Furthermore, the engineering stress shows a linear decrease, which indicates that the cross-section is reduced due to a constant speed of crack growth. Phase VI is the rupture of the specimen, which shows the spontaneous failure of the specimen in the crack tip under high local heat production, followed by a smooth cooling curve, as shown in Figure 2.

### 4.2. Neck Propagation/Cold-Drawing

As shown in Figure 2, for neck propagation (phase III, cold drawing), the engineering stress as well as the maximum temperature in the specimen are constant. This indicates that the strain in the unstretched as well as in the necked region stays also at a constant value in this deformation phase.

The present experiments show a correlation between the crosshead speed and the crystallinity, the drawing stress, and the deformation-induced heating. In Figure 4, the engineering stress and strain for cold-drawing (phase III in Figure 2) are given with respect to the crosshead speed.

Both stress and strain show for small crosshead speeds a high value, with 36.6 MPa engineering stress at a crosshead speed of 10 mm/min and a respective engineering strain of about 1.7%. With a crosshead speed of about 100 mm/min, there is a minimum in the stress and strain curves at 26.6 MPa stress and 1.2% strain, respectively. For higher crosshead speeds, there is a moderate increase in the drawing stress and strain until the stress values seems to stay constant for crosshead speeds higher than 300 mm/min with engineering values of about 30 MPa stress. Only the strain curve rises constantly up to 1.75% at 400 mm/min, but with a considerable standard deviation of about 0.23%. The remarkably high temperature in the transition zone also shows a dependency on the crosshead speed; see Figure 5 (left). For the evaluation, the maximum temperature in the temperature profile over the specimen is used, as one can see in Figure 3. For the smallest crosshead speed $v_{ch}=10\;\frac{mm}{min}$, the measured temperature is about 38 °C. This is near the theoretically predicted temperature for an adiabatic volume element of about 39.8 °C (Equation (11)) as only the dissipation of mechanical work in the transition zone is considered, with an drawing stress of $\sigma_{draw}=36.5\;MPa$, speed of neck propagation of $v_{neck}=12\frac{mm}{min}$, and a material property heat capacity of $c_{p}=1100\frac{J}{g}$ and density of $\rho =1.38\frac{g}{cm^{3}}$.

(11)$$\Delta T=\frac{\sigma_{draw}\;v_{ch}}{c_{p}\;\rho \;v_{neck}}$$

It is mentionable that the standard deviation of the temperature values is small, with the highest value of 1.6 K at a crosshead speed of 100 mm/min and 63 °C, respectively. Therefore, the error bars are not visible in the left diagram of Figure 5.

A similar degressive trend regarding the temperature can be observed for the draw ratio of the deformed material, as it is shown in Figure 5 (right) with respect to the crosshead speed, but with a much greater standard deviation of the temperature measurement. Additionally, there is an inflection at the crosshead speed of 100 mm/min, and the standard deviation is at its highest with a value of 0.5.

The temperature profile for cold-drawing shows great deviation for different crosshead speeds (see Figure 6). For lower speeds, there is a sharp peak at the beginning of the transition zone with a low maximum temperature, a small heated area, and a clear preheating of the undrawn material due to heat conduction. With higher speeds, the maximum temperature increases, and the shape of the peak in the temperature profile changes from sharp to a round appearance. The position of the maximum temperature chances into the stretched material. The temperature gradient of the undrawn material increases strongly.

### 4.3. DSC Measurement

For the cold-drawn specimens, DSC measurements were carried out at samples stretched with different crosshead speeds. DSC curves including the unstretched material are shown in Figure 7. As one can see, in comparison to the initial, undrawn material, the glass transition shifted from about 75 °C to a lower temperature of about 63 °C for the drawn material. Another point is that the distinct peak for cold crystallization as known for amorphous PET is not present. These points are seemingly without influence of the crosshead speed. Lastly, the melting peaks of the drawn material are getting narrower but at the same time higher and sharper with the increasing crosshead speed. The melting enthalpy for all the evaluated materials is more or less at about 53.5 J/g, in contrast to the melting enthalpy of the amorphous material with about 40.5 J/g respectively, as shown in Figure 8.

Additional to internal friction, strain-induced crystallization is reported as a source for the heating of the specimen to reach the temperature, as shown in Figure 5 and Figure 6 [8].

To quantify the latent heat through strain-induced crystallization $\Delta H_{strain}$, which should be released in the transition zone, the sum of the melting enthalpy $\Delta H_{melt}$ and the enthalpy between glass transition and the beginning of the crystallite melting $\Delta H_{T_{g}\ldots T_{melt}}$ is used in Equation (12):(12)$$\Delta H_{strain}=\Delta H_{melt}+\Delta H_{T_{g}\ldots T_{melt}}$$

The values of the enthalpy with the associated integration temperatures for different crosshead speeds are given in Table 1.

Figure 8 shows that the trend of the strain-induced crystallization $\Delta H_{strain}$ (given as squares) with respect to the drawing speed shows the same trend that is shown by the temperature and the drawing ratio. The determined enthalpy of the cold crystallization of the initial material (dotted line) is about −39 J/g, which is nearly equivalent to the melting enthalpy, and proves nearly amorphous condition. For the drawn material, the enthalpy of cold crystallization during DSC measurement (triangles) is much smaller than for the initial material and decreases for higher crosshead speeds, which indicates that there is a certain amount of crystallized material in the drawn material, and thus the calculation in Equation (12) is feasible.

### 4.4. Temperature Model

With the above-mentioned model, a temperature prediction is possible for different crosshead speeds. Figure 9 shows for the temperature profile for cold-drawing a comparison of the predicted temperature and the IR measurement for four exemplary crosshead speeds.

There is a good match for the model and the real data of the measurement in the area of the drawn material for higher crosshead speeds. For the crosshead speed of 25 mm/min, the match is not as good. The maximum temperature for the model is always greater than the real data and the shape of the peak is not reproduced correctly, especially for higher crosshead speeds. Furthermore, the area of unstretched material in front of the active zone is in reality more heated with respect to the model.

## 5. Discussion

The self-heating of PET during cold-drawing is not explainable just with the dissipation of mechanical work through viscous friction, as the predicted temperature in the transition zone should stay between 35 °C and 39 °C with Equation (10). As Liao et al. states, it is a combination of the effects of viscous friction and latent heat due to crystallization [12]. The diminishing increase to higher temperatures for faster crosshead speeds becomes plausible if one sees the increase in the draw ratio for a higher crosshead speed (Figure 5, right) that comes with a greater orientation of the molecular chains, which is also reported by different other authors [4,6,12,16]. This orientation, coupled with the energy from the heating due to viscous friction, seems to enable the material to crystallize faster and at lower temperatures than those of the amorphous, unstretched APET. This deduction is supported by the work of Bartolotta et al., who found similar effects in films of APET, which were cold-drawn at a crosshead speed of 2 mm/min. Subsequent annealing at different temperatures and durations showed crystallization at lower temperatures and at faster rates compared to undrawn material [17]. It is remarkable that there is an inflection in the trend of the draw ratio with respect to the crosshead speed at 100 mm/min, and the standard deviation is at its highest. At this crosshead speed, the resulting temperature in the transition zone equals the glass transition temperature of the stretched material at 63 °C, as derived from Figure 7.

In Figure 8, the increase in the resulting enthalpy with respect to the crosshead speed is shown, and one can see that the increase in the strain-induced resulting enthalpy slows down at a crosshead speed of about 200 mm/min to 300 mm/min. It seems that for this crosshead speed, the material is mostly oriented and crystallized, and an additional increase in crosshead speed brings little gain in this regard. Therefore, the strengthening of the material by an increase in the crystalline phase is mostly effectual up to a crosshead speed of about 300 mm/min. For an explanation of the decrease in drawing stress and strain for small crosshead speeds and the minimum at 100 mm/min, one has to look at the maximum temperature in dependence of the crosshead speed in Figure 4 (left). As stated above, the temperature rise is explained with heat dissipation by viscous friction and latent heat by crystallization. The similar trend for crystallization enthalpy and maximum temperature in the neck corresponds to the latent heat in the transition zone generated by crystallization. In this context, the latent heat due to strain-induced crystallization, which is determined through the DSC measurements in Figure 8, does not match for a low crosshead speed of 25 mm/min, if one considers that the maximum temperature of about 47 °C is mostly due to the dissipation of mechanical work, as Equation (10) suggests. The trend for stress and strain in Figure 4 seems to be a superposition of the weakening of the amorphous phase of the material and the strengthening effects due to orientation and crystallization. Therefore, the decrease in stress and strain and the following minimum is mostly caused by the temperature increase. In contrast, for higher crosshead speeds, the stress and strain trend is also affected by orientation and crystallization until the crystallization of the material seems to increase no further.

A three-phase model for the crystallization of cold-drawn PET in which there exists amorphous regions alongside crystallites and smectic areas in the cold-drawn material is proposed by Bartoletta et al. This idea was picked up by Asano et al. They added a nematic state as a preliminary stage for the smectic and crystallized material. In their work, they report the evolution and growth of the crystalline lamellae due to annealing [17,18]. It seems that for higher crosshead speeds with a higher resulting temperature, these annealing effects play a role in cold-drawing itself. As Figure 6 shows, the heated area is much wider for higher crosshead speeds. Considering the longer time to cool down as well as the higher orientation and chain mobility due to the higher temperature, the shift in the temperature maximum as shown in Figure 6 may be an effect of time-delayed crystallization because of the annealing effect, as proposed by Bartoletta. The temperature gradient to the undrawn material gets greater for high crosshead speeds not only due to the higher resulting temperatures, but also because of the high speed of the neck propagation and the poor heat conduction of PET. It is possible that the discrepancy between the measured temperature and determined strain-induced crystallization for the lower crosshead speeds is also because there are firstly nemantic areas and the crystallization occurs much later and slower.

Figure 8 shows only a moderate increase in the resulting enthalpy starting at a crosshead speed of about 100 mm/min. The resulting temperature for this crosshead speed coincides with the glass transition temperature of the drawn material of about 63 °C, as mentioned above. It may provide another piece of evidence for the annealing of the oriented material in the transition zone for higher speeds. This shift of the glass transition to lower temperatures is unexpected. A shift of the glass transition to higher temperatures was expected, as there is an increasing number of restrictions on the polymer chains due to orientation and crystallization. Other authors also report a decrease of the glass transition of about 12 K for cold-drawn PET, as shown in Figure 7. The reason is not yet clear, and studies of this phenomenon are not known. Anisotropic internal stress imposed by the cold drawing is an explanation that Bartoletta proposed [9,17,18]. Another point is that the DSC curves show no clear difference in the temperature region where cold crystallization should take place. It is interesting that the melting enthalpy of the cold-drawn material is greater than that for the amorphous material. This indicates that the cold-drawing enables a greater crystallization degree than the thermal crystallization of the amorphous material. Again, this can be caused by the annealing of the smectic areas, as Bartolotta et al. proposed.

It seems that the enthalpy relaxation peak at the glass transition is rather prominent in the cold drawn material, which was expected because of the confinements due to the high orientation while the material is cooling after the transition zone. For a crosshead speed of 50 mm/min, the enthalpy relaxation seems to be the greatest, with smaller peaks at higher speeds. This may be again due to the higher temperature in the transition zone with annealing effects, which results in lesser internal stress.

The temperature model reproduces the temperature data of the measurements only in the area of the stretched material in a satisfactory way. For the shape of the peak and the area of the unstretched material in front of the transition zone, the model only shows a correct trend. This is founded by the heat generation, which is for the model defined to only one volume element.

Regarding the findings in this work, the introduced thermomechanical model should be modified so that the time-delayed heating due to strain-induced crystallization can be addressed as well as the uneven energy dissipation over the transition zone. With this, the shift in the position of the maximum temperature in the temperature profile and the change in the peak shape, as shown in Figure 6, may be reproduced. To realize this, the different heat sources should be determined separately. One approach can be to determine the local viscous friction due to the deformation in the transition zone. Another approach may be the investigation of the crystallization kinetics in the transition zone. Minor changes to the model should be the acknowledgement of the changes in the heat capacity, density, and heat conduction due to the temperature change and the crystallization and orientation of the material.

## 6. Conclusions

Investigation of the drawing process of amorphous polyethylene terephthalate with IR thermography and optical strain measurement shows the typical trend for stress–strain curves for polymers with necking behavior, as well as for the local maximum temperature in the sample.The measured temperatures in the transition zone for cold-drawing exceed the strain-induced heating due to the dissipation of mechanical work through viscous friction. An additional heat source is strain-induced crystallization, which is dependent on the molecule orientation and mobility and seem to be time-dependent.The softening and hardening of the material with respect to the crosshead speed can be accounted to the superposition of the rising temperature in the transition zone, the higher orientation, and the increasing crystallinity of the material.For cold-drawn material, the glass transition is shifted to smaller temperatures. This shift of about 12 K in the case of PET is independent of the crosshead speed. The cause is yet unknown.Cold crystallization is not clearly differentiable for the cold-drawn material. If the enthalpy is integrated over a wide area from glass transition up to the melting, a dependency of the crosshead speed with the same trend as the increasing temperature and the draw ratio can be evaluated. This points out that some time-dependent molecular rearrangements and crystallization occurs.The melting enthalpy for cold-drawn material is greater than that for the amorphous material. This is accounted to the high orientation of the polymer chains, which results in the development of smectic areas in which crystallization is faster as well as possible at lower temperatures. The smectic areas enable the crystalline lamellae to grow in a time-delayed annealing process.The model for the temperature profile for cold-drawing, based on the first law of thermodynamics, gives good results for small crosshead speeds of 100 mm/min and lower. For higher crosshead speeds, there is a deviation in the peak obtained from the model with respect to the real data. For correction, the time-delayed annealing and crystallization processes should be considered.

## Figures and Tables

**Figure 1 polymers-11-01871-f001:**
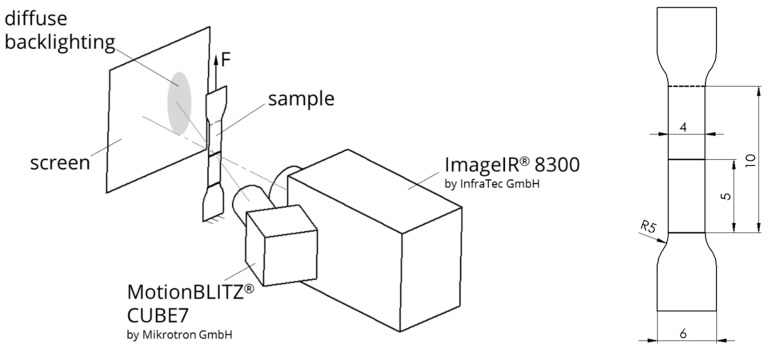
Experimental setup and modified specimen geometry.

**Figure 2 polymers-11-01871-f002:**
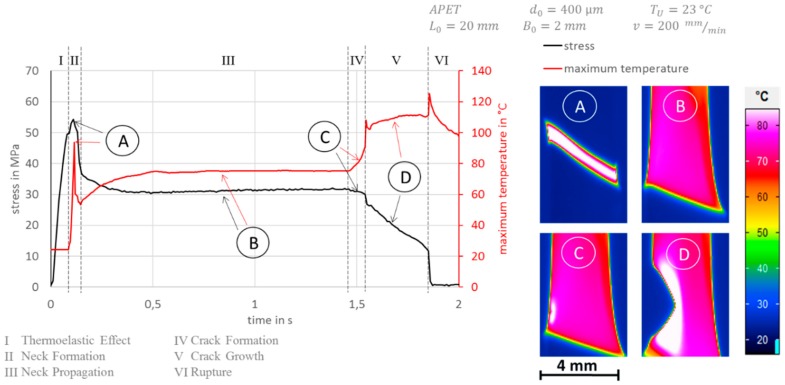
Left: Characteristic stress evolution (black) and maximum temperature evolution (red) with respect to time for amorphous polyethylene terephthalate (APET) in a uniaxial tensile test with the deformation phases I: elastic behavior, II: neck formation, III: neck propagation, IV: crack formation, V: crack growth, VI: rupture. Right: thermographic pictures A–D for corresponding deformation phases II–V.

**Figure 3 polymers-11-01871-f003:**
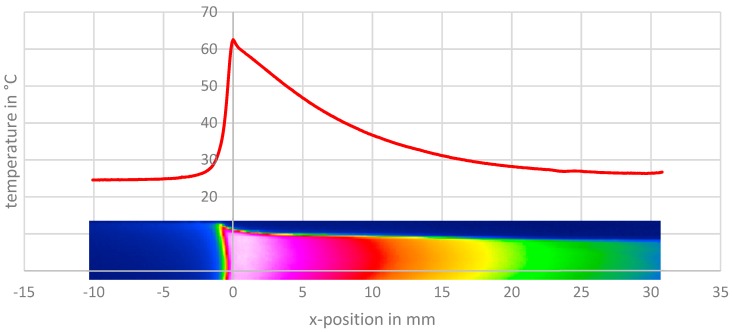
Temperature profile during neck propagation in the middle of a specimen of APET with a drawing speed of 100 mm/min.

**Figure 4 polymers-11-01871-f004:**
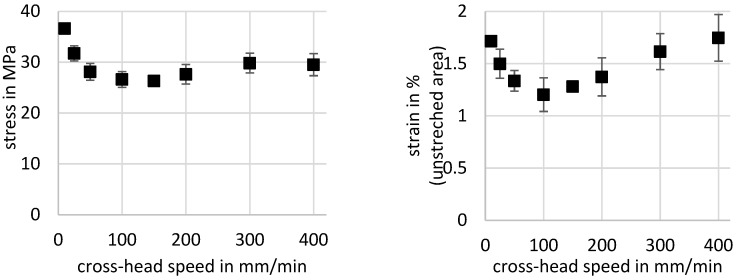
Engineering stress (left) and strain (right) against crosshead speed.

**Figure 5 polymers-11-01871-f005:**
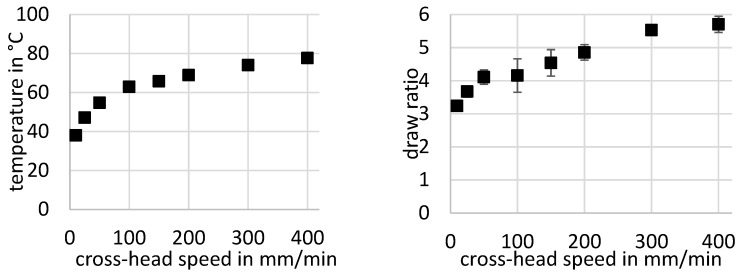
Maximum temperature in the transition zone (left) and draw ratio (right) against crosshead speed.

**Figure 6 polymers-11-01871-f006:**
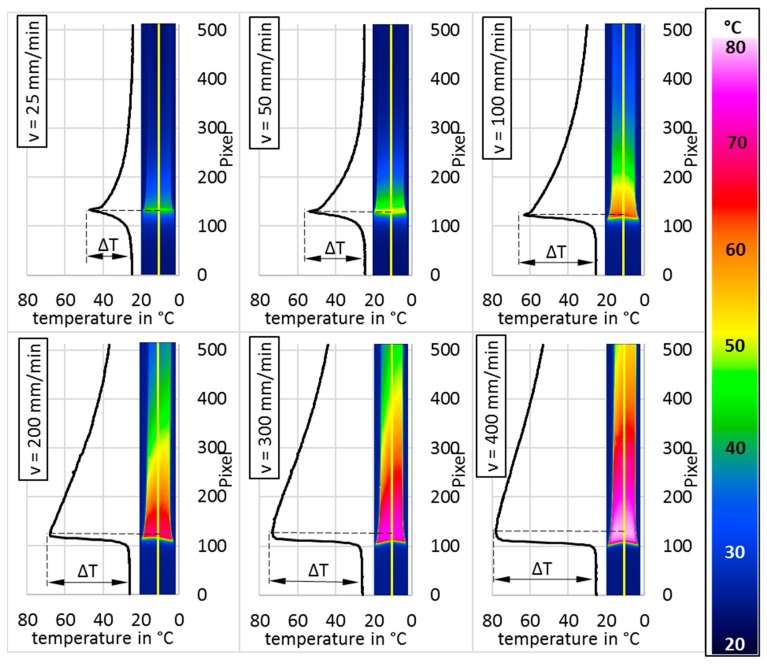
Temperature profile for cold-drawing at different crosshead speeds.

**Figure 7 polymers-11-01871-f007:**
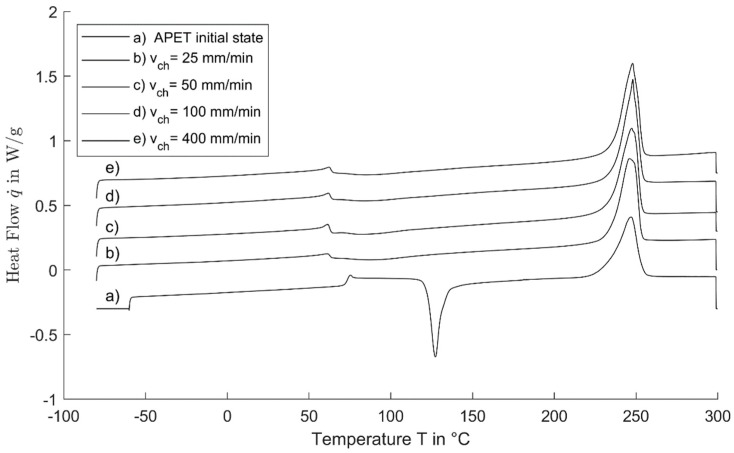
Differential scanning calorimetry (DSC) curves for APET, stretched with different crosshead speeds in comparison to the initial material; the curves are shifted with respect to each other.

**Figure 8 polymers-11-01871-f008:**
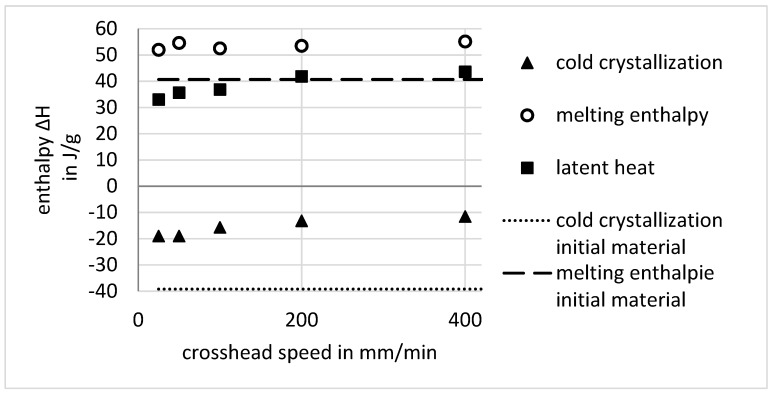
Strain-induced crystallization enthalpy *ΔH*, and cold crystallization $\Delta H_{T_{g}\ldots T_{melt}}$ against crosshead speed.

**Figure 9 polymers-11-01871-f009:**
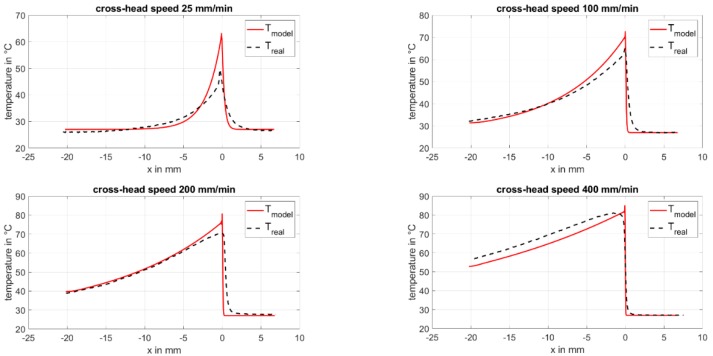
Temperature profile for cold-drawing as predicted with the numerical model and the real data of the measurements.

**Table 1 polymers-11-01871-t001:** Enthalpy values with integration temperatures.

Crosshead Speed	Integration Temperture	$\Delta H_{T_{g}\ldots T_{melt}}$	Integration Temperature	$\Delta H_{melt}$	$\Delta H_{strain}$
mm/min	°C	J/g	°C	J/g	J/g
25	63.2–212	−19	212–260	51.95	31.4
50	62.9–205	−19	205–260	54.6	34.1
100	62.6–208.7	−15.7	208.7–260	52.5	35.3
200	63.3–211.5	−13.2	211.5–260	53.5	38.8
400	64–209	−11.55	209–260	55.15	42.1

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
