# Peer review of "Thermomechanical Characterization and Modeling of Cold-Drawing of Poly(ethylene Terephthalate)"

_polymers, 2019, doi:10.3390/polym11111871_

Round 1
Reviewer 1 Report
Reviewer’s comments on the manuscript
Thermo-mechanical characterization and modelling of cold-drawing of poly(ethylene terephthalate)
This paper deals with the tensile deformation of PET up to the break-up. A particular interest is brought to deformation-induced heating of the sample: a model for the temperature evolution in cold drawing is given. This article is well built. However, the contribution of this paper is not clearly defined, perhaps because it does not refer to any article published after 2015.
Major remarks:
This study must be better situated in relation to the existing literature, in order to highlight its scientific contribution. For the authors, the term Q.ambience is a combination of convection and heat radiation. The equation (7) corresponds to Newton's law about convection, but how does it take into account the radiation, which is classically a function of the temperature in power 4? How are the error bars determined in Figures 4 and 5? • They are not visible in Figure 5a and they are very high in Figure 5b, between 100 and 200 mm / min where, in addition, an inflection appears. This inflection (Figs.5) is not discussed: why?
Minor remarks:
h is defined too far in the text, which makes difficult to understand the relationships (8 and 9) English can be improved as shown by the few examples below:"this model can describes ..." (line 60), "... there were to markings ..." (line 100), "These cooling curve ..." (line 163), …
Considering the above-points, I do not recommend acceptance of the paper. Major revisions are expected.

Reviewer 2 Report
This paper reports on the effect of strain rate on the measured temperature rise during the cold-drawing of PET. The temperature-sample location profile is modelled using an expression for ΔT that is derived in the introduction to the paper. While the paper is certainly interesting, it is not without its issues. These are described below.
The literature review element of the paper is too brief and the reader is left in a position where he/she is unsure on how the work described in this paper develops on what has been done before. For example, it is stated that Bazhenov ‘tried to give a mathematical description…’, but it is not made clear whether or not this approach was successful or how the model proposed in this paper relates to what has been done before (by Bazhenov).
Equation 8. The approach seems reasonable, but I would request that the authors consider the effect of the change in heat capacity as the amorphous PET crystallises. The difference may be insignificant, but it should at least be acknowledged. α, I interpret this as the thermal diffusivity, which would also change as the material becomes oriented and crystallinity develops. The authors should consider these comments and the implications relating to the assumption that they do not change during the cold-drawing process.
The APET sample. Could the thin film be already oriented? Does the sample contract on heating?
Why is the glass transition temperature of the APET greater than the oriented samples?
Does figure 7 show an indication of cold-crystallisation on heating above the glass transition?
Equation 12. It is not clear why (or how) DHstraincan be obtained from figure 7. This should be justified more thoroughly and the analytical approach specified.
Figure 9. The legends in the figures are not displayed fully so its is not clear which line is measured and which is predicted.
Round 2
Reviewer 1 Report
This paper deals with the tensile deformation of PET up to the break-up. A particular interest is brought to deformation-induced heating of the sample: a model for the temperature evolution in cold drawing is given. This article is well built.
The contribution of this paper, relative to those of previous works reported in literature, has been clarified in the corrected version. The paper has been thoroughly corrected.
So, considering the improvement of the corrected paper, I recommend its acceptance.